# Alpha-Synuclein FRET Biosensors Reveal Early Alpha-Synuclein Aggregation in the Endoplasmic Reticulum

**DOI:** 10.3390/life10080147

**Published:** 2020-08-11

**Authors:** Fabiana Miraglia, Verdiana Valvano, Lucia Rota, Cristina Di Primio, Valentina Quercioli, Laura Betti, Gino Giannaccini, Antonino Cattaneo, Emanuela Colla

**Affiliations:** 1Bio@SNS Laboratory, Scuola Normale Superiore, 56126 Pisa, Italy; miragliafabiana@gmail.com (F.M.); verdiana.valvano@gmail.com (V.V.); lucia.rota@sns.it (L.R.); cristina.diprimio@sns.it (C.D.P.); valentina.quercioli@sns.it (V.Q.); antonino.cattaneo@sns.it (A.C.); 2Department of Pharmacy, University of Pisa, 56126 Pisa, Italy; laura.betti@unipi.it (L.B.); gino.giannaccini@unipi.it (G.G.); 3Neurotrophins and Neurodegenerative Diseases Laboratory, Rita Levi-Montalcini European Brain Research Institute, 00161 Rome, Italy

**Keywords:** alpha-synuclein, oligomers, aggregation, endoplasmic reticulum, FRET, biosensors, Parkinson’s Disease, alpha-synucleinopathy.

## Abstract

Endoplasmic reticulum (ER) dysfunction is important for alpha-synuclein (αS) acquired toxicity. When targeted to the ER in SH-SY5Y cells, transient or stable expression of αS resulted in the formation of compact αS-positive structures in a small subpopulation of cells, resembling αS inclusions. Thus, because of the limitations of immunofluorescence, we developed a set of αS FRET biosensors (AFBs) able to track αS conformation in cells. In native conditions, expression in i36, a stable cell line for ER αS, of intermolecular AFBs, reporters in which CFP or YFP has been fused with the C-terminal of αS (αS-CFP/αS-YFP), resulted in no Förster resonance energy transfer (FRET), whereas expression of the intramolecular AFB, a probe obtained by fusing YFP and CFP with αS N- or C- termini (YFP-αS-CFP), showed a positive FRET signal. These data confirmed that αS has a predominantly globular, monomeric conformation in native conditions. Differently, under pro-aggregating conditions, the intermolecular AFB was able to sense significantly formation of αS oligomers, when AFB was expressed in the ER rather than ubiquitously, suggesting that the ER can favor changes in αS conformation when aggregation is stimulated. These results show the potential of AFBs as a new, valuable tool to track αS conformational changes in vivo.

## 1. Introduction

In the last decade, the necessity to investigate alpha-synuclein (αS) conformational changes in vivo and the pathological consequences of its misfolding on cell function, gave rise to the development of fluorescent molecular tools, able to track the formation of αS high molecular weight (HMW) species [1,2,3,4,5]. αS, the dominant component of Lewy bodies, proteinaceous inclusions found in Parkinson’s disease (PD) patients, can aggregate in highly ordered protofibrils, according to a nucleation reaction, with different intermediate states of oligomer assembly [6]. Specifically, at least two types of oligomer structures have been described in vitro [7,8,9], while for protofibrils a growing consensus based on cryo-electron microscopy (cryo-EM) data showed a Greek key type of conformation [10,11,12], rich in parallel β-sheets, orderly stacked to form a filament, although other arrangements have been described [13]. To this variety of structures and shapes corresponds different degrees of toxicity and a different ability to propagate [14]. However, much of structural data on αS conformation derived from studies on in vitro preformed αS HMW species. Förster resonance energy transfer (FRET), which is based on the efficient transfer of excitation energy between two fluorophores if sufficiently close, is a quantitative method to study proteins interaction and conformation in vivo [15]. Development of FRET reporters based on the fusion of αS with YFP and/or CFP provided insights on αS native conformation in cells and its ability to propagate [1,16]. With this in mind, in the present work, we developed a set of αS FRET biosensors (AFBs) with the aim to detect a variety of αS structures in cells and we used them to investigate αS’s link to the endoplasmic reticulum (ER), a cellular organelle that has been shown to be affected by αS toxicity [17]. Alterations of the ER-Golgi transport [18,19,20,21] and from Golgi to endosomes/lysosomes [22,23,24] concomitant with the induction of ER stress and accumulation of toxic αS species associated with the ER/Golgi membrane, implicated the ER dysfunction as an important step in α-synucleinopathy [25,26]. In addition, the high affinity of αS to bind phospholipids and biological membranes [14,27], its subcellular association with synaptic vesicles and the ER/Golgi are all evidence indicating that the ER could be an early site of aggregation [17,28]. In agreement with this, comparison of the ability to FRET of AFBs when expressed in the ER or ubiquitously in native or pro-aggregating conditions, suggests that αS compartmentalization in the ER can favor formation of HMW species. We believe that through analysis of AFBs behavior it is possible to detect αS conformational changes and to gain a valuable insight in the native or aggregated structures of αS as they occur in cells.

## 2. Materials and Methods

### 2.1. Molecular Cloning

To target αS into the ER, wild-type (WT) αS was amplified from the pcDNA 3.1-αS plasmid, already present in the lab, using the primers PstI-αS-Forw (5′-GGGCTGCAGATGGATGTATTCATGAAAGGA-3′) and XhoI-αS-Rev (5′-ACGCTCGAGAAGGCTTCAGGTTCGTA-3′) and cloned into pCMV-myc-ER using the PstI and XhoI restriction sites, in frame with an ER signal (ERS) and a retention sequence (SEKDEL) at 5′ or 3′, respectively, of αS. In order to establish stable cell lines expressing αS into the ER, ER-αS-myc from pCMV-myc-ER was subcloned into pcDNA4/TO (Life Technologies, Carlsbad, CA, USA), by restrictions with HindIII and XbaI.

To clone the intermolecular FRET probes, CFP or YFP from the vectors pE-CFP-C1 and pE-YFP-C1 (Life Technology) were amplified using the primers NotI-CFP-Forw (5′-ATAAGAATGCGGCCGCAGAACAAATGGTGAGCAAG-3′) and XbaI-CFP-Rev (5′-GGATCTAGATTATTATCTAGATCCGGTGGATCCCGG-3′) and subcloned into the NotI restriction site, at the 3′ of WT αS in pcDNA3.1(+)-WT-αS or in pCMV-myc-ER WT αS. In both constructs, a poly-linker of 10 amino acids was inserted between αS and the fluorescent proteins.

For the intramolecular αS FRET bionsensor, YFP from pEYFP-C1 vector was subcloned within NheI and HindIII sites into the N-terminal of αS-CFP in pcDNA 3.1. This time, a 5 or 10 amino acids poly-linker was added between YFP or CFP and αS.

The positive FRET control, already available in the lab, is a vector that contains CFP and YFP fused together in frame within the same construct (pECFP-EYFP). The resulting fusion protein CFP-YFP is always able to generate a FRET signal because of the vicinity of the two fluorophores.

For the ER negative control, CFP and YFP were separately amplified, using the NotI-CFP-Forw and XbaI-CFP-Rev primers, as above indicated. CFP and YFP were then subcloned in two distinct pCMV-myc-ER vectors (pCMV-ER-CFP and pCMV-ER-YFP). Co-expression of both plasmids in cells should not lead to a FRET signal as the two proteins do not interact.

### 2.2. Cell Cultures, Transfection and Stimulation with Pro-Aggregating Agents

In order to establish tetracycline-inducible cell lines expressing αS in the ER, neuroblastoma SH-SY5Y cells, already present in the lab, were first transfected with pcDNA 6/TR (Life Technologies, Carlsbad, CA, USA) and then with pcDNA4TO-ER-αS.

Through antibiotic resistance, several clones were isolated and amplified. Line i36 was then selected for further analysis and routinely cultured in DMEM medium (Sigma-Aldrich, St. Luis, MO, USA) supplemented with 10% (*v*/*v*) fetal bovine serum (FBS) (Euroclone, Milan, Italy), 1% (*v*/*v*) glutamine, 100 U/m penicillin and 100 mg/mL streptomycin, 3 µg/mL, Blasticidin and 50 µg/mL Zeocin™ (ThermoFisher Scientific, Eugene, OR, USA). Expression of αS was induced by addition of 0.5 µg/mL doxycycline. For cell toxicity assay, 5 × 10^4^ cells were plated in triplicates in 24 wells, induced with doxycycline and differentiated with 10 µM retinoic acid. Toxicity was determined as a function of Trypan Blue (Sigma-Aldrich, St. Luis, MO, USA) exclusion assay after 15 days from induction.

For transient transfection, 2.5 × 10^4^ SH-SY5Y cells, routinely cultured in complete medium (DMEM, 10% (*v*/*v*) FBS, 1% (*v*/*v*) glutamine, 100 U/m penicillin and 100 mg/mL streptomycin), were plated in 8-well glass slide chamber (Thermofisher) and transfected with either WT ER-αS in pCMV vector or WT αS in pcDNA3.1. After 24 h the medium was replaced and the cells were differentiated with 10 µM retinoic acid. Cells were fixed for immunofluorescence after 2 days from differentiation.

For FRET experiments, the day before transfection, cells were seeded at 2 × 10^5^ cells in 0.01 mg/mL poly-D-lysine coated glass bottom WillCo-dishes^®^ (WillCo Wells B.V., Amsterdam, The Netherlands) to be 70–90% at transfection and induced with 0.5 µg/mL doxycycline. The following day, cells were transfected with the appropriate biosensor. After 24 h, the medium was replaced and pro-aggregating agents were added for 24 h. Agents included 100 µM 6-hydroxydopamine (6OHDA); 100 µM leupeptin or 1 mg/mL microsomes-associated αS aggregates obtained from the spinal cord of diseased A53T αS transgenic (Tg) mice [29], or a combination of leupeptin and microsomes-associated αS aggregates in concentration as above. The following day, cells were either fixed for FRET imaging or harvested for western blot (WB).

### 2.3. Cell Fractionation, Western Blot and Immunofluorescence

For immunoblotting, cells were lysed according to protocols previously described [25,26,29,30]. Briefly, cells were lysed in cold lysis buffer containing 1% *v*/*v* Triton-X 100 (Tx), with protease and phosphatase inhibitor cocktails (PhosSTOP™ and cOmplete™, Mini, EDTA-free kits, Merck, Darmstadt, Germany) in phosphate-buffered saline (PBS). In order to obtain ionic (SDS) and non-ionic (Tx) detergent soluble fractions, cell lysates were centrifuged at 16,000× *g* for 15 minutes at 4 °C. The supernatants were transferred into new Eppendorf tubes and labelled as Triton-X 100 soluble fractions (Tx-S). The pellets were washed twice with lysis buffer and dissolved in 100 µL of lysis buffer supplemented with 1% *v*/*v* SDS, sonicated and boiled at 95 °C for 5 minutes. These pellets were labelled as Triton-X 100 insoluble fractions (Tx-I). Protein amount of cell lysates was determined using the Pierce BCA Protein Assay Kit (Thermofisher).

For western blot analysis, 5 µg of Tx-S and 10 µg of Tx-I cell fractions were loaded on a 4–20% Criterion™ TGX™ Precast Protein Gel (Bio-Rad, Hercules, CA, USA) and transferred onto nitrocellulose membrane (Bio-Rad). Transfer efficiency was verified by Ponceau S staining (Sigma Aldrich). Membranes were then blocked for 30 minutes with 5% *w*/*v* non-fat dry milk (Bio-Rad) in PBS buffer containing 0.01% *v*/*v* Tween-20 (PBS-T) at room temperature and then incubated in specific primary antibody dissolved in 2.5% *w*/*v* non-fat dry milk in PBS-T, O/N at 4 °C. After washing and incubation with the appropriate secondary antibody, the chemiluminescent signals were visualized with SuperSignal West Femto Maximum Sensitivity Substrate (ThermoFisher) using a CCD-based Bio-Rad Molecular Imager ChemiDoc System. Band intensities were quantified using Quantity One software (Bio-Rad), normalized against Ponceau S staining.

For cell imaging, transfected cells were fixed for 15 minutes at room temperature with 3.7% paraformaldehyde in PBS as previously described [29,30] or with 2% paraformaldehyde for FRET experiments.

Primary antibodies were as follows: mouse Syn1, clone 42, 1:5000 (BD Biosciences, NJ, USA); mouse Myc 1:1000 (9B11, Cell Signaling Technology, Leiden, The Netherlands), rabbit polyclonal Grp94 1:1000 (ab13509, Abcam, Cambridge, UK), and rabbit polyclonal GFP (A-11122, Thermofisher) 1:2000. Secondary antibodies were as follows: for WB, anti-mouse-IgG BP-HRP (sc-516102) and anti-rabbit IgG-HRP (sc-2357, Santa Cruz Biotechnology, Inc, Santa Cruz, CA, USA) 1:3000; for IF Alexa Fluor^®^ 488 anti-mouse (A11001) and Alexa Fluor^®^ 555 anti-rabbit (A27039) 1:1000 (Thermofisher).

### 2.4. Microscopy and FRET Experiments

Images were acquired with the TCS SP2 laser-scanning confocal microscope (Leica Microsystems, Wetzlar, Germany) using a 63 x/1.4 NA HCX PL APO oil immersion objective. Fixed cells were maintained at room temperature. An Argon laser was used for CFP (λ = 458 nm) and YFP (λ = 514 nm), a He-Ne laser for Alexa Fluor 633 (λ = 633 nm). Acceptor photobleaching (AP) experiments were performed on fixed cells prepared as described in Section 2.3. First, the beam path settings were optimized using the positive and negative control samples. The 512 × 512 pixels images in the CFP channel (458nm ex/470–500 nm em) and YFP channel (514 ex/530–600 nm em) were acquired before and after acceptor photobleaching at 400 Hz, line average 1 with a 4.00 x zoom. When the photobleaching configuration was set, the entire cell body was selected (typically corresponding to an average area of 430 ± 140 μm^2^) and bleached at 514 nm with 40% power laser. FRET efficiency was calculated by the FRET Wizard Leica Confocal Software according to the formula:FRET = (D_Post_ − D_Pre_)/D_Post_(1)
where D_Post_ is the fluorescence intensity of the donor after acceptor photobleaching, and D_Pre_ the fluorescence intensity of the donor before acceptor photobleaching. About 30 circular ROIs (3.15 μm^2^) inside each cell were selected after acceptor photobleaching to quantify FRET efficiency in different cellular compartments.

### 2.5. Statistical Analysis

FRET experiments were performed in triplicate. In each experiment, ten cells per sample were analyzed. FRET efficiency values are expressed as percentage of Equation (1). The distribution of data was tested for normality by the Shapiro–Wilk test. Data showing a skewed distribution were described by box plot, where the box shows the lower (25th) and upper (75th) percentile, the line inside the box indicates the median (50th percentile), the whiskers show the minimum and maximum values excluding outliers. Circles, triangles and rhombi outside the boxes represent outliers. Differences between medians were determined through one-way ANOVA, followed by Dunn’s multiple comparisons test using GraphPad Prism (GraphPad Software Inc., La Jolla, CA, USA). For the time course expression of ER αS after doxycycline administration and for the transfection efficiency experiment, data analysis was performed using one-way ANOVA, followed by Tukey’s multiple comparisons test. For the cell toxicity assay, data analysis was performed using the Student *t*-test.

## 3. Results

### 3.1. Formation of αS Positive Inclusions in SH-SY5Y Cells Expressing αS in the ER

In order to investigate if the ER could favor αS propensity to aggregate, SH-SY5Y cells were transiently transfected with WT αS, targeted either ubiquitously or to the ER. Compared to regular expression in the cytosol (Figure 1A), transfection of ER-αS WT, resulted in the accumulation of small, compact αS-positive structures, that co-localized with the ER marker Grp94, in a small subpopulation of cells (about 15%) (Figure 1).

Because of the scarcity of cells bearing compact αS structures, we decided to generate an inducible cell line that stably expresses WT αS-ER after addition of doxycycline.

Analysis of Triton soluble (Tx-S) and insoluble (Tx-I) lysates of i36 cell line showed a clear “on/off” shift after induction with doxycycline displaying accumulation of αS in both fractions which lasted with similar intensity over several days of induction (Figure 1C,D). Immunofluorescence analysis confirmed the presence of αS-positive structures in about 20–25% of cells, although no clear sign of cellular toxicity was associated with prolonged induction of ER-αS or with formation of ER-αS inclusions (Figure 1E,F). Thus, expression of αS in the ER can result in the formation of compact αS structures that resemble protein inclusions.

### 3.2. Development of Alpha-Synuclein FRET Biosensors (AFBs)

While previous data showed that expression of αS in the ER could be a suitable strategy to study the impact of compartmentalization and membrane association on αS aggregation in cells, such models are inadequate for quantitative predictions to establish whether the ER, as opposed to the cytoplasm, favors formation of αS HMW species. Thus, to bypass this issue and gain insight into the structure of αS in cells, we developed a set of αS FRET-based biosensors (AFBs) to track αS species. Initially, our aim was to design reporters capable of detecting αS only when folded in an oligomer or an aggregate type of assembly, ranging from a dimer to a multimer. Thus, CFP and YFP were added specifically at the C-terminal of two distinctive αS cDNAs, respectively (αS-CFP and αS-YFP), leaving the N-terminal of αS in both molecules available to make interactions. In addition, a poly-linker of 10 amino acids between αS and the fluorescent proteins was also included to increase probes flexibility. Co-expression of the single probes ubiquitously in cells (αS-CFP and αS-YFP in Figure 2A) or targeted to the ER (αS-CFP ER and αS-YFP ER in Figure 2B) allows tracking of αS aggregation as emission of the FRET signal with this biosensor is only possible when two or more αS monomers are close enough that the two fluorophores can efficiently transfer energy, according to FRET principles.

Differently, in order to investigate the conformational behavior of the single αS monomer, we cloned YFP and CFP at N-terminal and C-terminal, respectively, of the same αS molecule (intra-AFB, YFP-αS-CFP in Figure 2D). As opposed to previous biosensors, intra-AFB is sensitive to changes within the αS monomer such as the switch from a closed type of conformation, where the N-terminal and C-terminal of αS are at short distance and generate a FRET signal, to a more open structure of αS, where the N-terminal and C-terminal move away and the FRET signal is lost [31,32]. Such a transition has been described to correspond to an initiation of aggregation, that brings the N- and C- termini of different YFP-αS-CFP molecules in close proximity [33]. Single FRET probes were expressed in SH-SY5Y cells to validate their expression and their distribution in the ER. As shown in Figure 2C, compared to αS alone or to AFB ER, the ubiquitous probes for inter-AFB showed a more homogeneous distribution in the entire cell, including the expression in the nucleus, whereas the ER probes for inter-AFB ER displayed a different profile that merged significantly with Grp94 localization, suggesting that AFB ER was correctly expressed mainly in the ER.

### 3.3. Inter-AFBs Expression in i36 Reveals Early αS Aggregation in the ER under Stress Stimuli

In order to investigate whether the ER relocation of αS may affect the protein propensity to aggregate, doxycycline-induced i36 cells were transfected with either inter-AFB or inter-AFB ER and stimulated with several pro-aggregating agents. In order to obtain aggregation in cells, we used a wide range of compounds, previously described to stimulated αS fibrillization. These agents included 6-OHDA, a well-known neurotoxin associated with formation of αS toxic adducts and oligomers [34] and with selective degeneration of dopaminergic neurons in acute PD animal models [35]; leupeptin, a potent lysosomal inhibitor, an organelle implicated in αS degradation [36,37] and microsomes-associated αS aggregates, αS toxic species purified from diseased Prp A53T Tg mice able to induce formation of endogenous αS inclusions [29,30].

After 24 h of treatment, cells were fixed and imaged for AP FRET while correct expression of both biosensors in native and pro-aggregating conditions were confirmed by WB (Figure 3C and Figure 4C). As shown in Figure 3 and Figure 4, i36 cells transfected with either inter-AFB or inter-AFB ER did not show a significant FRET signal in untreated conditions compared to the negative control, a reporter generated by the co-expression of CFP and YFP cloned in two separate vectors. This indicates that in physiological conditions and in all the cellular compartments analyzed, inter-AFBs cannot track native αS conformation including the membrane-bound multimeric structure first described by Burré et al. [38].

In contrast, when i36 cells transfected with inter-AFB ER were treated with pro-aggregating agents, this reporter generated a positive FRET signal (Figure 3A,B).

Specifically, while all the pro-aggregating agents induced FRET, the strongest value was obtained when cells were treated with leupeptin, confirming that αS is a substrate for lysosomal degradation and that inhibition of this pathway is directly involved in generation of αS HMW species [24,39,40,41,42]. Interestingly, microsomes-associated αS aggregates or 6-OHDA were not as efficient as leupeptin in inducing αS aggregation, whereas when cells were treated with a combination of leupeptin and microsomes-associated αS aggregates, this treatment did not show a cumulative effect. In this case, inter-AFB ER signal was significantly diminished compared to leupeptin alone, suggesting that these two treatments may compete for AFB availability during stimulation of protein interactions. Nevertheless, these data showed that inter-AFB expressed in the ER is efficiently sensitive to pro-aggregating conditions and able to track αS HMW species in cells. On the contrary, when AFB was expressed ubiquitously in i36 cells, the corresponding FRET signal under pro-aggregating stimuli was significant only when compared to the negative control but not when compared to AFB in untreated conditions (Figure 4A,B).

Thus, while ubiquitous inter-AFB still retains the ability to respond to pro-aggregating stimuli, when it is expressed in the ER, its efficiency to sense αS conformational changes greatly increases, suggesting that the ER may favor αS oligomer formation in the early stages.

Notably, to see if such interactions were specific for αS or they could be achieved with any protein, a negative control, in which CFP and YFP were cloned in two separate vectors and targeted to the ER, was also used (Ctrl–ER, Figure 3A,B). Significantly, cells expressing AFB ER and treated with pro-aggregating agents showed areas with increased positive FRET signal compared to the ER-targeted negative control, indicating that mere expression of fluorescent proteins in the ER is not sufficient to generate FRET. Thus, the propensity of αS to aggregate in the ER seems to be an intrinsic and specific property of the protein per se.

### 3.4. Intra-AFBs Expression in i36 Confirms αS Monomeric Conformation in Physiological Conditions

In order to get a better insight in the structure of αS in native conditions, we repeated the above experiment and transfected induced i36 cells with the intra-AFB, a biosensor able to track conformational changes within the single αS molecule. As opposed to what was observed in the case of inter-AFBs, intra-AFB showed a significant FRET signal in basal conditions when compared to the negative control, suggesting the presence in cells of mainly αS monomer with a compact, globular conformation, where YFP and CFP at the N- and C-termini, respectively, of the biosensor are in close proximity and therefore able to generate a FRET signal (Figure 5A,B). Interestingly, differences in the FRET signal between intra-AFB and inter-AFBs under normal conditions were not due to a variability in the transfection efficiency of the biosensors, which instead showed a similar percentage of cells expressing the reporters in i36 cells (Appendix A).

Furthermore, when i36 cells were transiently transfected with intra-AFB and treated with similar pro-aggregating agents as above, FRET values remained comparable to untreated conditions. Thus, the lack of increased FRET signal in pro-aggregating conditions suggests that in our settings this biosensor cannot discriminate between native and aggregated αS and therefore it is more suitable to study αS intramolecular interactions in physiological conditions.

## 4. Discussion

Increasing amounts of evidence suggest that overexpression of αS plays a central role in PD pathogenesis with the ER possibly being one of the most susceptible subcellular compartments for the development of αS pathology [18,25,26,29,30,43,44]. According to these findings, we decided to evaluate the behavior of αS when directly expressed in the ER and we developed a set of FRET-based biosensors able to sense variation of the αS structure in native and pro-aggregating conditions. Because of the intrinsic limitations of the FRET technique and because of the unnatural tagging of αS, such as adding large fluorescent proteins especially at N-terminal, which can affect the protein behavior, we cannot conclude at this time that AFBs have the same properties as untagged, endogenous αS or that AFBs can detect all αS possible conformations in normal and pro-aggregating conditions. We tried to overcome this issue by adding poly-linker amino acid sequences between αS and the fluorophores, as described by Nakamura et al. 2009 [1]. Interestingly in their paper, CD spectra of untagged αS and αS-CFP or YFP-αS-YFP were similar, and presented similar conformational transition upon addiction of artificial membranes. Thus, through complementary studies on these biosensors, we were able to detect in cells early changes in αS conformation underlying the aggregation process.

Depending on salt concentrations, pH or other interactors in the cellular environment, the αS monomer has a closed and globular conformation, where the N- and C- termini are close to each other and shield the NAC core from hydrophobic interactions [45]. According to our findings, intra-AFB is able to track this closed structure, where the YFP and CFP fluorophores located at N- and C-termini of αS, respectively, are close enough to generate a positive FRET signal (<2–10 nm) in untreated conditions (Figure 6A).

This is in line with what has been reported by others using similar intramolecular αS FRET biosensors [1]. In addition, expression of both inter-AFBs did not result in FRET, confirming that the αS monomer is the prevalent species in basal conditions. However, under pro-aggregating stimuli, the persistent invariance of the FRET signal generated by the intra-AFB compared to control conditions, suggested that intra-AFB can sense not only the αS native conformation, but also αS HMW species. Nevertheless, because the FRET value of intra-AFB does not differ much in untreated conditions or after pro-aggregating stimulation, intra-AFB does not seem adequately efficient in tracking aggregation-related changes in αS structure and thus it appears to be more suitable to study αS in its native state. In addition, we cannot exclude that the intra-AFB can detect membrane-bound αS, as it has been previously described [38], although such species have been estimated to account only for 10–15% of the total αS pool. Here upon binding to liposomes, αS can multimerize in an antiparallel configuration through the interaction of the α-helices at the N-terminal of two distinct molecules. Interestingly, the intramolecular FRET reporter presented by Nakamura et al., loses FRET signal upon addition of acidic phospholipids, indicating that while this biosensor can still sense αS bound to membranes, the resulting positive FRET signal is the result of only the monomeric protein. More studies will be necessary to understand if our intra-AFB can appreciate and distinguish the variety of αS structures in native conditions.

Differently from above, following pro-aggregating stimulation, the inter-AFBs are swiftly able to track formation of αS oligomers and protofibrils (Figure 6B). αS oligomers, enriched in β-sheets secondary structures, have been described to undergo conformational conversion and turn from an antiparallel to a parallel orientation in protofibrils [8,13,46]. Besides a broad range of sizes, toxic oligomers have been described to have a cylindrical, barrel-like or a doughnut shape, with a diameter ranging from 90 to 100 Å and a central core of about 25 Å [8]. In addition, Cryo-EM studies at 3.7 Å resolution, described β-sheets parallel oriented protofibrils with a 5 nm diameter that aggregate in fibrils with a 10 nm diameter [10,12], in line with those described for PD or multiple system atrophy (MSA) patients [47]. The sizes of these oligomers and protofibrils structures fall within the FRET measurable range therefore it might be possible that AFBs are able to sense formation of these structures. Thus, inter-AFBs, through the interaction of their parallel oriented fluorescent probes (αS-CFP and αS-YFP) at the αS C-termini, can efficiently detect the interaction of two or more αS molecules generating the positive FRET signal observed in our experiments.

Furthermore, although we cannot confirm that the αS species detected by inter-AFBs are indeed pathogenic, the AFBs sensed a distinct variation in αS structure that was not present in native conditions. Interestingly, inter-AFBs were differently sensitive to the stimulus given, indicating that, although we used widely known agents described to foster αS aggregation [29,30,35,36], not all the treatments impacted the formation of αS toxic species in a similar way.

More importantly, formation of αS HMW species was favored when the inter-AFB was targeted to the ER. Although αS was initially identified at the presynaptic terminals as a synaptic vesicle-binding protein [28], multiple evidence indicate that αS associates and/or accumulates within the ER. First, αS can bind to BIP/grp78, a luminal component of the ER translocon in physiological conditions [26,48]. Second, αS is a substrate of misfolding-associated protein secretion (MAPS) system, an unconventional secretion process associated with the cytosolic side of the ER membrane [49]. In addition, the ER retention 1 (RER 1) protein, an ER factor able to promote retention/retrieval of immature proteins from the Golgi to the ER, has been shown to promote αS degradation via the proteasome system [50]. Finally, induction of ER stress and disruption of the ER-Golgi-membrane vesicles traffic has been widely associated with toxic αS in cells, primary neurons, yeast and mice [18,21,26,51,52]. In agreement with these data, we have shown how formation of ER-associated αS oligomers precedes ER stress and α-synucleinopathy in a mouse model of PD [25,26] and how these HMW αS species can self-propagate and be extremely neurotoxic in primary neurons [30]. In addition, αS aggregates appear to be particularly detrimental to the ER membrane, as opposed to the αS monomer, since they can bind to SERCA, an ER Ca^2+^ pump and induce directly Ca^2+^ release in the cytosol [53]. Finally, high-resolution imaging found that αS-immunopositive LBs/LNs from human PD brains are crowded with a membranous environment, including vesicles and dysmorphic organelles [54].

Taken together, these results confirm the tight association between αS and the ER and reinforce the hypothesis that the ER plays a crucial role in fostering formation of initial toxic species of αS, as it was shown by the increased sensitivity of our AFBs ER in detecting αS oligomers under stress conditions. Finally, the data obtained in this work highlight the power of AFBs as investigative tools to provide complementary information on tracking native and aggregated αS species.

## 5. Conclusions

In summary, our work has established a new set of tools that allows to track changes in αS structure as it happens in live cells. Through the comparison of the properties of both AFBs we were able to gain insight in the folding pattern of αS and to study early stages of aggregation. We believe that AFBs could therefore be a powerful strategy for the screening of potential therapeutic compounds against αS aggregation.

## Figures and Tables

**Figure 1 life-10-00147-f001:**
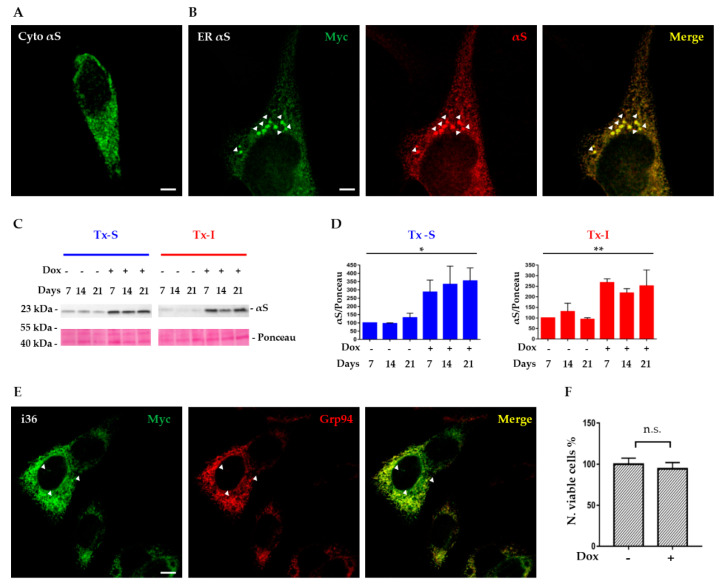
Expression of alpha-synuclein (αS) in the endoplasmic reticulum (ER) leads to formation spherical structures resembling aggregates. Immunofluorescence analysis of wild-type (WT) αS transiently expressed in the ER of SH-SY5Y cell line showed spherical structures resembling aggregates that co-localized with the ER marker grp94. SH-SY5Y cells were transfected with ER-WT αS in pCMV-ER (**B**) or WT αS in pcDNA3.1 (**A**). In fixed cells, αS was detected with incubation with Myc antibody. (**C**) Time course analysis of i36 cell lines, stably expressing αS in the ER under the induction of 0.5 µg/mL doxycycline (Dox). Cells were lysed at different intervals and Tx-S and Tx-I fractions were isolated. Cell lysates were then run on a SDS-PAGE and immunoblotted with Syn-1 antibody. The experiment was repeated three times with comparable results. (**D**) Quantitative analysis of Tx-S (blue) and Tx-I (red) fractions of i36 cells with/out induction of doxycycline revealed a gradual increase of αS expression after induction in both fractions. Values on the graph represent a ratio between αS expression normalized by the Ponceau staining and are the mean ± SEM (n = 3). * (*p* < 0.05); ** (*p* < 0.01). (**E**) Co-localization of myc-tagged ER αS in i36 cell line with the ER marker, Grp94. Immunofluorescence images (**A**,**B**,**E**) were acquired with a Leica confocal microscope SP2, using a 63 x objective. Scale bar, 10 μm. (**F**) Survival of the i36 clone after two weeks of induction with doxycycline. Cells were seeded in triplicates in a 24-well plate and stimulated for the expression of ER αS with doxycycline. Viable cells were determined with the Trypan Blue exclusion assay. Values on the graph are expressed as % of control and represent the mean ± SEM (n = 3). n.s., not significant.

**Figure 2 life-10-00147-f002:**
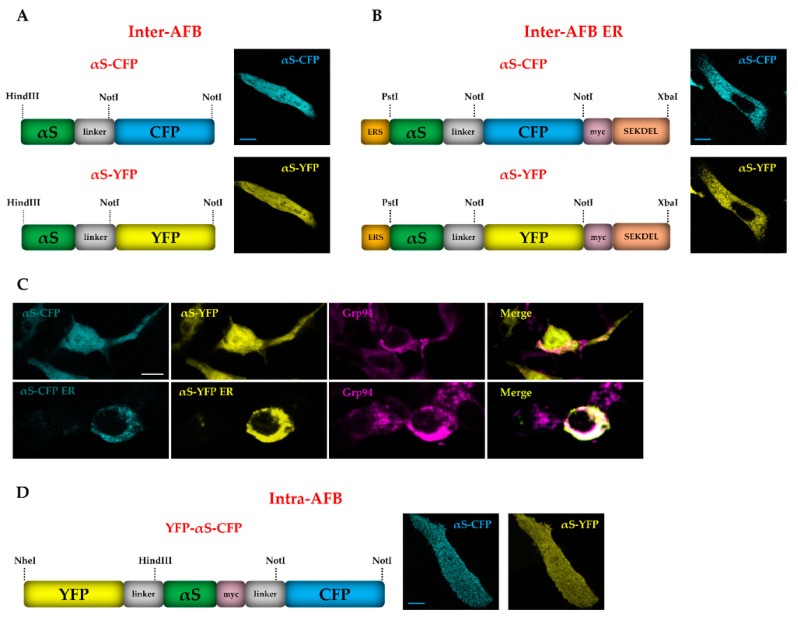
Schematic representation of αS Förster resonance energy transfer (FRET) biosensors (AFBs) vector maps and expression of single probes in SH-SY5Y cells. Schematic and immunofluorescence images of single probes of (**A**) the ubiquitous intermolecular biosensor, inter-AFB, where CFP and YFP were cloned separately at the C-terminal of WT αS (αS-CFP and αS-YFP) in pDNA3.1; (**B**) intermolecular biosensor target to the ER, inter-AFB ER, where CFP and YFP were cloned separately at the C-terminal of ER αS in pCMV-ER (ER αS-CFP and ER αS-YFP), containing an ER signal (ERS) at the N-terminal of αS and an ER retention sequence (SEKDEL) at the C-terminal of the single probes; (**D**) intramolecular fluorescent reporter (intra-AFB) where YFP and CFP were cloned at the N and C termini of the same αS cDNA in a pcDNA 3.1 vector. (**C**) Co-localization of inter-AFB with Grp94, showed a homogeneous distribution in the cells different from what was observed for inter-AFB ER, which localizes in the ER as confirmed by overlapping staining of Grp94. Confocal images of fixed cells expressing AFBs were acquired in the CFP (Ex = 458 nm) and YFP (Ex = 514 nm) channels with Leica SP2 confocal microscope, using a 63x objective. Scale bar, 10 μm.

**Figure 3 life-10-00147-f003:**
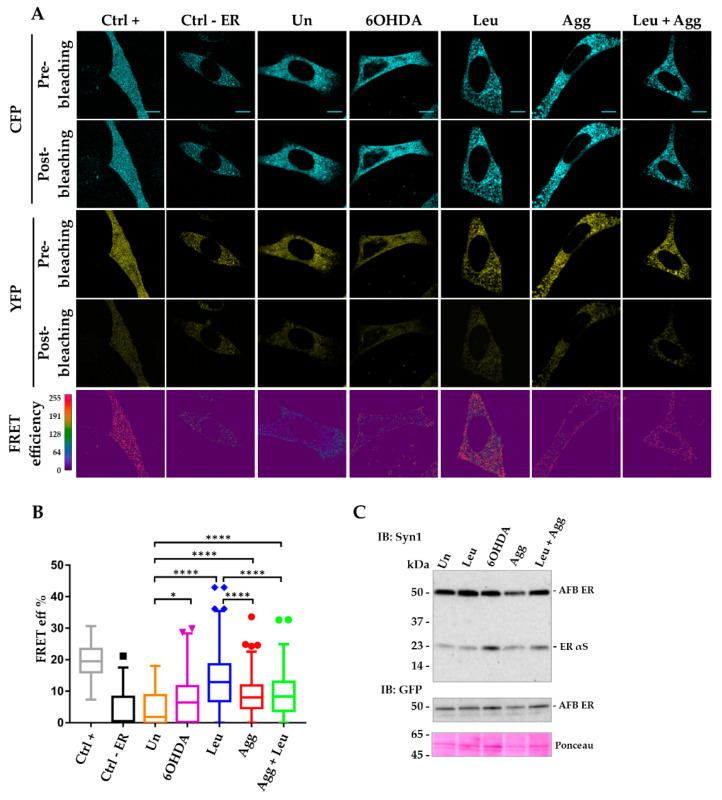
FRET analysis of inter-AFB ER expressed in i36 cell line highlights early αS structural variation under pro-aggregating conditions. Doxycycline-induced i36 cells were transiently transfected with inter ER AFB or positive (Ctrl +) and negative (Ctrl-ER) controls. The day after cells were treated with pro-aggregating stimuli (Leupeptin, Leu; 6-hydroxydopamine, 6OHDA; microsomes-associated αS aggregates, Agg) and fixed for FRET analysis or lysed for western blot (WB) after 24 h. (**A**) FRET images were acquired in the donor (CFP) and acceptor (YFP) channels, before and after photobleaching (pre- and post-bleaching) the acceptor. FRET efficiency, calculated by the FRET Wizard Leica Confocal Software, showed regions of higher FRET values (red regions) in all samples treated with stimuli compared to untreated conditions and Ctrl-. Scale bar, 10 μm. (**B**) Box plot of FRET efficiency of samples in A expressed as percentage were obtained using the formula (Dpost-Dpre)/Dpost. Each box plot shows the median (50th percentile), values to the 1.5 interquartile range (whiskers), 25th to 75th percentile range (box). Circles, triangles and rhombi outside the boxes represent outliers. One-way ANOVA followed by Dunn’s multiple comparison test. * (*p* < 0.05); **** (*p* < 0.0001). (**C**) WB analysis of Tx-S fraction of cell lysates transfected with inter-AFB ER in untreated and treated conditions. Immunoblotting with Syn1 or GFP antibody indicated the correct expression of the inter-AFB ER at the expected size of about 50 kDa in stimulated and control conditions.

**Figure 4 life-10-00147-f004:**
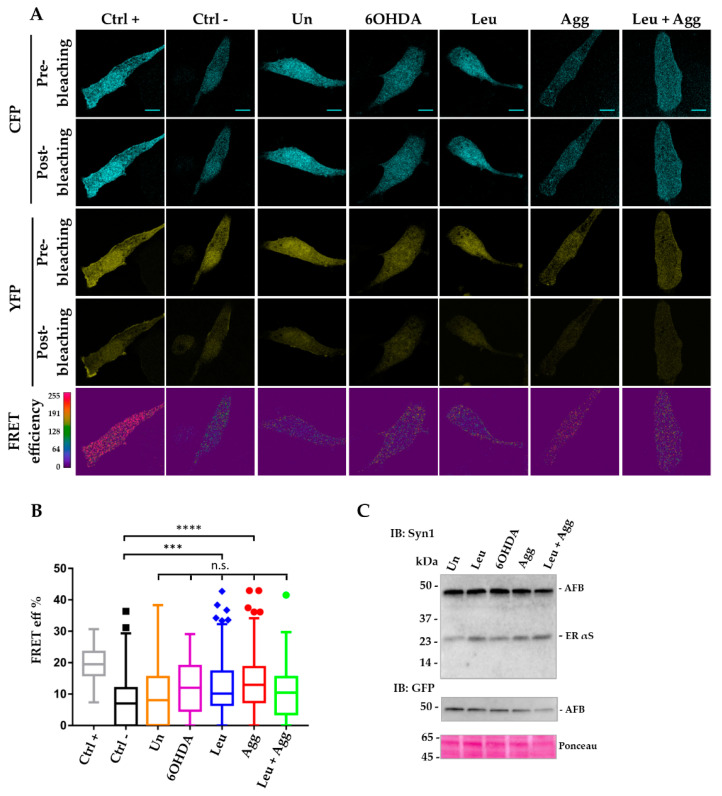
FRET analysis of ubiquitous inter-AFB expressed in i36 cells is not efficient in detecting αS aggregation under normal or pro-aggregating conditions. Induced i36 cells were transfected and assayed following the same protocol as described in Figure 3. (**A**) Images of AP FRET efficiency showed high FRET values (red regions) for the Ctrl + and a distribution of lower FRET values (blue and green regions) for all the other conditions comparable to the Ctrl–Scale bar, 10 μm. (**B**) Box plot of FRET efficiency indicated that only samples treated with leupeptin or microsome-associated αS aggregates show significant values compared to Ctrl–but not when compared to inter-AFB in untreated condition. *** (*p* < 0.001); **** (*p* < 0.0001); n.s., not significant. (**C**) WB analysis of Tx-S fraction of cell lysates transfected with inter-AFB in untreated conditions or treated with pro-aggregating agents. Immunoblotting analysis with Syn-1 or GFP antibody of Tx-S fractions of i36 cell lines transfected with inter-AFB shows the correct expression of the biosensor with the predicted molecular weight of about 45kDa in treated and control conditions.

**Figure 5 life-10-00147-f005:**
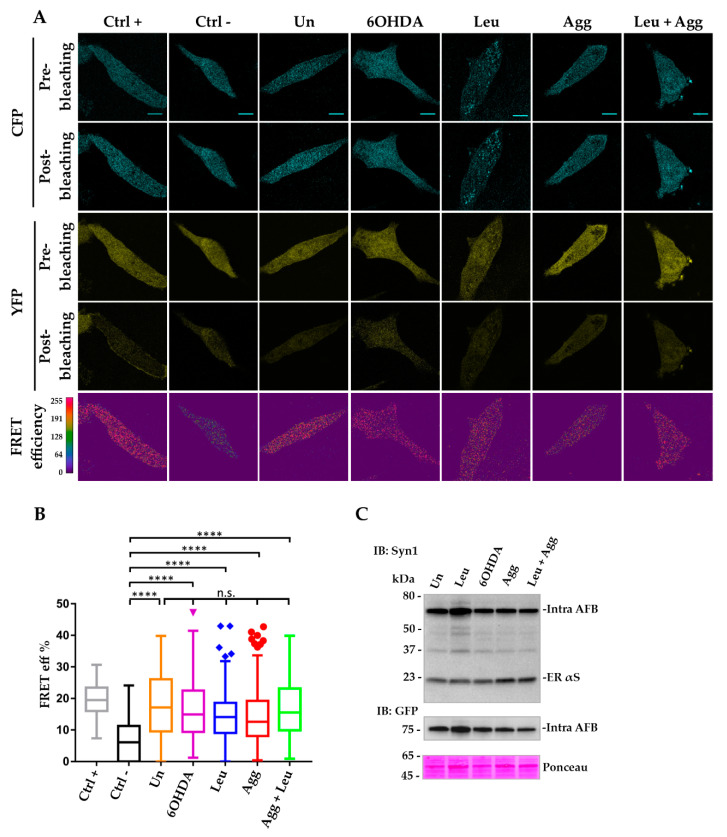
FRET analysis of intra-AFB expressed in i36 shows that this reporter is sensitive to αS native structure but not to αS HMW species in pro-aggregating conditions. Induced i36 cells were transfected and assayed following the same protocol as described in Figure 4. (**A**) Images of AP FRET efficiency showed high FRET values (red regions) for the Ctrl + and for intra-AFB samples in untreated and treated conditions when compared to the Ctrl –. Scale bar, 10 μm. (**B**) Box Plot of FRET efficiency indicating that intra-AFB shows similar FRET values in untreated and treated conditions significantly different from the Ctrl –, but cannot distinguish between normal or stimulated states suggesting that intra-AFB is only suitable to sense native αS conformation. **** (*p* < 0.0001); n.s., not significant. (**C**) WB analysis of Tx-S fraction of cell lysates transfected with intra-AFB in untreated conditions or treated with pro-aggregating agents. Immunoblotting analysis with Syn-1 or GFP antibody of Tx-S fractions of i36 cell lines transfected with intra-AFB shows correct expression of the biosensor with the predicted molecular weight of about 75 kDa confirming the correct fusion of the two fluorescent probes to the same monomeric αS molecule.

**Figure 6 life-10-00147-f006:**
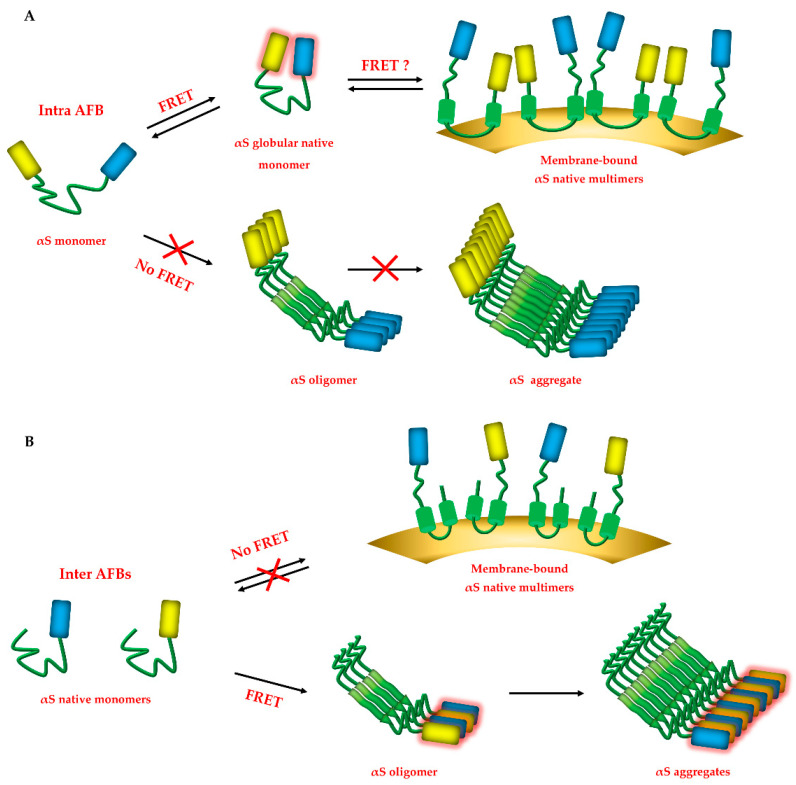
Schematic panels describing AFBs hypothetical behavior in detecting different αS species in the cellular environment. (**A**) Intra-AFB could, theoretically, sense the native state of αS (globular monomeric or the bidirectional conversion to a membrane-bound multimeric, as shown by the double arrows) as well as aggregated forms. In our system, though, Intra-AFB is sensitive only to the native state of αS, since when aggregation is stimulated no significant variation of FRET signal for the intra-AFB is recorded (crossed arrow). (**B**) Inter-AFB can sense only HMW species under pro-aggregating conditions, at a distance of 2–10 nm, according to FRET principles, but cannot detect native multimeric αS structures bound to membranes as described by Burrè et al. [38], probably because the fluorophores are too distant.

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
