# Peer review of "Alpha-Synuclein FRET Biosensors Reveal Early Alpha-Synuclein Aggregation in the Endoplasmic Reticulum"

_life, 2020, doi:10.3390/life10080147_

Round 1
Reviewer 1 Report
Alpha-synuclein FRET biosensors reveal early alpha-synuclein aggregation in the endoplasmic reticulum
In this study, Miraglia et al. developed a set of alpha-synuclein FRET biosensors (AFBs) with the intention of tracking its conformation and oligomerization dynamics inside living cells. The community should appreciate the effort made to venture into the in vivo characteristics of alpha-synuclein, where the authors show that when looking specifically at the ER, pro aggregation stress generates FRET-positive oligomers of alpha-synuclein. Though there is plenty of studies regarding FRET and alpha-synuclein, many of them are made in vitro, so this is a welcome strategy that, with proper follow up studies, will shed light on the mechanisms of alpha-synuclein oligomerization and aggregation.
That being said, I have minor comments before the paper can be published:
If at all possible, it would be interesting to do an immunogold labelling for electron microscopy of the ER-related oligomers. If successful, these images would show the identity of the aggregates the authors describe in figure 4. Are they membranous-rich? Maybe lipofuscin-rich? I refer specifically to an example presented in figure 2 of the paper published by Dettmer et al. in Human Molecular Genetics, 2017.
Additionally, things that could be better explained in the paper include:
What is the reason for using different poly-linker lengths in the intramolecular alpha-synuclein FRET biosensor (line 83 in the methods)?
Experiment of figure 2. Why is there a 2 day time point only for the i11 clone? Either include a 2 day time point for i36 as well or get rid of it.
Why are the FRET events discussed in figure 6 (intra-AFB) assumed to be only monomeric ones? Couldn’t it be that two or more molecules of tagged alpha-synuclein are interacting in an antiparallel fashion that does not depend on the presence of pro-aggregation molecules?
How can the authors tell that the early processes observed are on-pathway to fibril formation? You don’t see increased toxicity (Figure S1). Either do a ThT staining or soften the statement.
A positive FRET event shouldn't be correlated with protofibril or fibrils solved by cryo-EM formation. Even under ideal conditions, the position of both N and C-terminal regions of alpha-synuclein is too dynamic and fails to give a strong enough electron density in these experiments. Equating the observations given here by FRET and the ones given by cryo-EM is a bit speculative.
Figures 1 and 2 feel preparative. I’d argue that they’re better suited for supplementary material.
Reviewer 2 Report
The article describes new FRET probes that can be used in combination to dissect the state of aS protein (monomer, oligomeric) in cellular context. A general concern of mine is what is the effect of YFP and CFP constructs (of significant size compared to aS) on aS conformation and what is the structural relevance of these aS species, native or aggregated, to physiological or pathogenic conditions respectively. But this is an inherent problem with this approach so I am not suggesting technical revisions on this point, rather it is appreciated that authors have also implied such limitations of the approach (e.g. Line 405).
Line 65: it is debatable whether experiments in cell lines are considered in vivo, I suggest to replace "in vivo" by "in cells" or similar.
Page 3, Lines 85-88: I found the description of FRET controls, negative control especially, obscure, please re-word or elaborate.
Line 127: Describes use of Ponceau S but provider missing. Line 134 describes that band intensities of Western blots normalized to Ponceau S staining. The accuracy of this type of normalization is questionable as in some cases (e.g. Figure 2A) the bands being barely visible. Also, would be good to indicate the size (in kDa), for the bands shown in the Ponceau strips for all applicable figures.
Line 125 describes that 5 μg of Tx-S and 10 μg of Tx-I were loaded, I suppose that protein concentration was determined before Western blot analysis? Please add a sentence to describe how was done.
Lines 138-140, please add complete information on the antibodies used including clone number and species and same for secondary antibodies.
Line 172: Is AT a typo instead of A53T?
Page 5, line 184 and figure panel D: I don't understand the comment that aS is human. Isn't this expected since the SH-SY5Y cell line is human derived?
Line 255: Spell out AP for acceptor photobleaching
Page 9, Figure 4: Spell out acronyms (u.a., Leu, Agg) in the figure legend
Line 306: How is the effect of 6OHDA on ER-aS Tx-I levels being interpreted?
Page 13, Figure 6D: Graph showing Tx-I is identical to Tx-I graph of Figure 5D. Indeed the quantitation shown in 6D does not match to the Western blot, at least visually. Please place the correct graph in panel D.
Page 13, Figure 6D: How is the effect of 6OHDA on Intra-AFB Tx-I levels (assuming there will be a difference after quantitation) being interpreted?
Line 381: FRET value of intra-AFB not being proportional to the amount of oligomers formed. Please provide evidence that oligomers are formed in these experiments with intra-AFB in i36 cell line.
Line 400: More recent publications about the structure of aS aggregates derived from disease brains (e.g. Schweighauser et al Nature 2020) worth to be mentioned as well.
Reviewer 3 Report
Overall assessment
The manuscript by Miraglia et al. is presenting interesting and novel tool to study the nature of alpha-synuclein in different cellular compartments, namely Endoplasmic reticulum (ER). They show that it is possible to express alpha-synuclein that translocate to ER (ER-AS), or remain in cytosol (Cyto-AS). Introducing the wt alpha-synuclein or the familial A53T variant to ER results in compact structures not seen when expressing cytosolic alpha-synuclein. It is a bit unclear why the authors introduce the A53T since it is not shown to accumulate more on ER and they do not include the mutant in further studies. They make two stable cell line with inducible alpha-synuclein that translocate to ER, however they only uses line i36, so i11 could be removed since it does not add to the data and conclusions. In addition, they make five different FRET biosensors sets (it is written a bit confusing, but my interpretation is as follows)
- YFP or CFP (termed Ctrl+)
- YFP or CFP with ER localization sequence (termed Ctrl - ER)
- Alpha-synuclein linked to either YFP or CFP (termed Inter-AFB)
- Alpha-synuclein linked to either YFP or CFP with ER localization sequence (termed Inter-AFB ER)
- Alpha-synuclein N-terminally linked to YFP and C-terminally linked to CFP (termed Intra-AFB)
Then Miraglia et al. transiently express the FRET biosensors (inter AFB and Inter AFB ER) in the inducible cell line i36 to investigate whether ER translocation affects the aggregation propensity of alpha-synuclein. The aggregation process is attempted to be stimulated by “pro-aggregation drugs”, however these are not described neither what they are, nor in which concentration they are used. The methods describe that the FRET efficiency is quantified in 30 ROIs in each cell in different cellular compartments, however the quantification of FRET efficiency does not reveal how many cells quantified in each experiment or how many biological replicates the data set contains?
Figure 4, 5, and 6 are very similar and differs only by what the cells are transfect with, namely Inter AFB ER in figure 4, Inter AFB in Figure 5, and Intra AFB. It is my opinion that combining these 3 figures a comparison of the differences between the biosensors are easier to follow (perhaps by putting the CFP and YFP panels in supplemental). Furthermore, on the panels of FRET efficiency, it is impossible to see the heat map, but if the scale is the same on all images, it can be shown just once with a note in the legend that it applies to all.
In the discussion make a nice cartoon, showing the information that can be extracted from use of the different FRET biosensors, however I don’t understand the bidirectional arrows, an especially don’t understand the interconnected arrow e.g. in A when AS aggregate. I suggest to make a crossed arrow to stress that the FRET signal is abolished.
The discussion highlights the reason for conducting this study, but it lack a discussion of whether attaching one or two large and highly folded proteins to alpha-synuclein, and in addition ER translocation sequence, affects the nature of alpha-synuclein. It appears that the triton insoluble material contain both the AFB and ER-alpha-synuclein, but there is no discussion on how that affects the FRET efficiency. I am also thinking whether the AFB are able to generate ER aggregates in wt SH-SY5Y cells, or if untagged ER alpha-synuclein is necessary?
Major concerns
- Aggregation of alpha-synuclein is shown by fractionation of cell extracts in triton soluble and Triton insoluble in fig 2,4, 5, and 6. It appears that the expression analysis on western blot is conducted once? It is of good scientific conductance to at least show that the data is reproducible, so the analysis of expression needs to be reproduced.
- Pro-aggregation drugs is not described at all. Studying aggregation is not trivial so modulation by an anti-aggregation drug e.g. like baicalein, EGCG, CLR01, the peptide ASI-1D, or similar, would add to the impact of this study.
- The number of biological replicates in not mentioned or the number analyzed transfected cells per experiment.
Minor concerns
- Fig 2 – line 194, and 195: 0,5µg/mL should be 0.5 µg/mL
- Concerning using ponceau staining as loading control; Are you normalizing to entire lane of the sample or a specific area e.g. as shown in figures 2, 4, 5, and 6?
Reviewer 4 Report
The manuscript by Miraglia and co-authors describes the use of novel FRET biosensors to check alpha-synuclein aggregation in the endoplasmic reticulum in neuronal-differentiated SH-SY5Y and i36 cells. Although the subject is of potential interest, there are several points that needs to be addressed to render the manuscript suitable for publication:
- FRET is largely dependent on changes in transfection efficiency. Did authors perform pilot investigation for assessing the optimal transfection efficiency to be used? These should be included.
- If this method is intended to be used for detecting alpha-synuclein aggregates in the ER as occurring in Parkinson’s disease pathophysiology it should be sufficiently sensible to detect it without the need of using a plasmid driving alpha-synuclein expression in the ER. The use of this strategy (directly targeting elevated levels of alpha-synuclein in the ER) should be justified, otherwise detecting alpha-synuclein aggregates in this district in cells where the protein is targeted there, sounds like detection of an artifact and limits the translational implication of developing these biosensors.
- The resolution of images in figure 2 looks very poor and the green signal looks overexposed.
- Figure 6: the YFP intensity seems to be very low in some of the conditions (e.g. Ctr -. 6OHDA, Agg, and Leu + Agg). The images should be replaced to have a more uniform intensity.
- - Page 14, Lines 343 to 345 "In addition, according to oligomers and fibrils structures recently published [10–12], tethering of different intra-AFBs species, that would bring together CFP and YFP from two different molecules, could theoretically still generate a positive FRET signal". According to the Cryo-EM data of Guerrero-Ferrera et al., 2018 (Figure 1 and 2) and Li et al., 2018 (Figure 3), the cross-section of the fibril illustrating the arrangement of the two protofilaments showed that the N-term or both the protofibrils are far from the C-term so it is difficult to obtain a FRET signal by using intra-AFB. For the same reasons, even though the distance between the fibril planes is only 4.9 A, the orientation of the fibrils makes the interaction between N-term and C-term very difficult.
Round 2
Reviewer 2 Report
All my previous comments are addressed. Thank you for revising the manuscript.
Author Response
Reviewer 2:
All my previous comments are addressed. Thank you for revising the manuscript.
Thank you for your comment.
Reviewer 3 Report
Minor concerns
- in ln 95 the toxicity assay is described to be conducted in 24-well plates but in legend ln 194 it is described as conducted in 96-well plate
- ln 367 there is a typo, poli-linker -> poly-linker
- In ln 372-377, there is a discussion of the structure of monomeric inter AFP as a closed globular structure, but no mention/discussion of the impact of the highly folded CFP or YFP.
Author Response
Reviewer 3:
Minor concerns
- in ln 95 the toxicity assay is described to be conducted in 24-well plates but in legend ln 194 it is described as conducted in 96-well plate.
Thank you for pointing out this discrepancy. The toxicity assay was carried out in 24-well and it has been corrected. Line 194.
2. ln 367 there is a typo, poli-linker -> poly-linker
We corrected that misspelling.Line 367.
3. In ln 372-377, there is a discussion of the structure of monomeric inter AFP as a closed globular structure, but no mention/discussion of the impact of the highly folded CFP or YFP.
We integrated the reviewer’s observation in line 363-370. We had already mentioned that unnatural tagging of alpha-synuclein can affect the protein structure and that this is a limitation of this technique. Nevertheless, we added specifically the reference to large fluorescent proteins to clarify further what type of tags are more impactful on alpha-synuclein behavior.
Reviewer 4 Report
Authors have addressed all the major issues and the manuscript may be suitable for publication though I still am not really convinced about a wide applicability of the method as it relies on the driven overexpression of alpha-synuclein in the ER. Will look forward optimization of it.
Author Response
Reviewer 4:
Authors have addressed all the major issues and the manuscript may be suitable for publication though I still am not really convinced about a wide applicability of the method as it relies on the driven overexpression of alpha-synuclein in the ER. Will look forward optimization of it.
Thank you for your comment and interests in the optimization of this method. We look forward of implementing this method in other cell models suitable for alpha-synuclein aggregation.